# A Broad-Spectrum Phage Endolysin (LysCP28) Able to Remove Biofilms and Inactivate *Clostridium perfringens* Strains

**DOI:** 10.3390/foods12020411

**Published:** 2023-01-15

**Authors:** Rui Lu, Banhong Liu, Liting Wu, Hongduo Bao, Pilar García, Yongjuan Wang, Yan Zhou, Hui Zhang

**Affiliations:** 1School of Food and Biological Engineering, Jiangsu University, Zhenjiang 212013, China; 2Jiangsu Key Laboratory for Food Quality and Safety-State Key Laboratory Cultivation Base of MOST, Jiangsu Academy of Agricultural Sciences, Nanjing 210014, China; 3School of Food and Biological Engineering, Shaanxi University of Science and Technology, Xi’an 710021, China; 4Dairy Research Institute of Asturias (IPLA-CSIC), 33300 Villaviciosa, Spain; 5Jiangsu Agri-Animal Husbandry Vocational College, Taizhou 225300, China

**Keywords:** *Clostridium perfringens*, phage, endolysin, biofilm, antimicrobial, duck meat

## Abstract

*Clostridium perfringens* is a gram-positive, anaerobic, spore-forming bacterium capable of producing four major toxins which cause disease symptoms and pathogenesis in humans and animals. *C. perfringens* strains carrying enterotoxins can cause food poisoning in humans and are associated with meat consumption. An endolysin, named LysCP28, is encoded by orf28 from *C. perfringens* bacteriophage BG3P. This protein has an N-terminal glycosyl–hydrolase domain (lysozyme) and a C-terminal SH3 domain. Purified LysCP28 (38.8 kDa) exhibited a broad spectrum of lytic activity against *C. perfringens* strains (77 of 96 or 80.21%), including A, B, C, and D types, isolated from different sources. Moreover, LysCP28 (10 μg/mL) showed high antimicrobial activity and was able to lyse 2 × 10^7^ CFU/mL *C. perfringens* ATCC 13124 and *C. perfringens* J21 (animal origin) within 2 h. Necessary due to this pathogenic bacterium’s ability to form biofilms, LysCP28 (18.7 μg/mL) was successfully evaluated as an antibiofilm agent in both biofilm removal and formation inhibition. Finally, to confirm the efficacy of LysCP28 in a food matrix, duck meat was contaminated with *C. perfringens* and treated with endolysin (100 µg/mL and 50 µg/mL), which reduced viable bacteria by 3.2 and 3.08 units-log, respectively, in 48 h at 4 °C. Overall, the endolysin LysCP28 could potentially be used as a biopreservative to reduce *C. perfringens* contamination during food processing.

## 1. Introduction

*Clostridium perfringens* is commonly present in soil, sewage, and the gastrointestinal tract of humans and animals [1], and it is the cause of severe infections such as food poisoning, necrotic enteritis, gas gangrene, and nonfoodborne gastrointestinal infections [2,3]. It is highly toxic due to its ability to produce at least 17 different toxins [4]. Among them, five types of toxins (A, B, C, D, or E) provide the basis for a typing classification system. Today, *C. perfringens* is one of the most prevalent causes of foodborne illness. In some cases, such as when keeping food at room temperature, *C. perfringens* can grow, multiply, and produce diarrhea-causing toxins. Additionally, transmission of *C. perfringens* type A from chicken carcasses is a major source of foodborne disease [5]. According to the Centers for Disease Control and Prevention, *C. perfringens* causes nearly one million illnesses annually in the United States and more than four million cases worldwide [6]. Overall, it is the second most important foodborne pathogen in developed countries [7].

Besides its ability to synthetize toxins, its persistence in the environment benefits from the capacity to form biofilms, which induces bacterial tolerance to disinfection [8]. Biofilms formed by *C. perfringens* could shield cells from atmospheric oxygen and high levels of antibiotics and coccidiostat [9]. In another related development, the biofilm formed by *C. perfringens* also protects cells from exposing hydrogen peroxide (10 mM) even though this bacterium is catalase-negative [10].

Although chemical preservatives (nitrite, phosphate, and organic acids) have been used to prevent food poisoning, some of them could have negative impacts on human health and can also affect the taste and quality of the product. Other alternatives are being explored, among them the biopreservation of food with healthy compounds. An example widely studied is the use of nisin. This bacteriocin secreted by *Lactococcus lactis* is active against gram-positive bacteria, including *Clostridium* spp. Nisin has been confirmed as a food preservative by the EU (E234), the World Health Organization (WHO), and the US Food and Drug Administration (FDA). Unfortunately, little research has evaluated the antibacterial activity of nisin against *C. perfringens* [11,12].

Other strategies for biocontrol of pathogenic bacteria include the use of bacteriophages (or phage). Phages and their lytic proteins have emerged as natural biopreservatives for the inhibition and eradication of bacterial contamination in a wide range of processing environments. Phages are safe to use against pathogens contaminating food, but regulations on their use as preservatives are limited to some countries [13]. A potential alternative to whole phages is the use of phage-encoded proteins that hydrolyze peptidoglycan (endolysins). Endolysins are expressed during the final stage of infection [14], but they are active against peptidoglycan when they are added externally to bacteria. Moreover, endolysins have some advantages compared to the use of phages, such as wider species specificity and the high unlikelihood of developing bacterial resistance [15,16].

To date, several endolysins encoded by *C. perfringens* phages have been identified and characterized, including those encoded by phages phi3626 [17], phiSM101 [18], phiCPS2 [19], and phiCP39O and phiCP26F [20], but few reports described their use as preservatives in food [21].

A *C. perfringens* phage, vB_CpeS_BG3P, was previously isolated in our laboratory from the sewage of a chicken farm and identified as belonging to the *Siphoviridae* family. Phage vB_CpeS_BG3P showed inhibition of a broad host range against different *C. perfringens* strains [22]. The objective of this study was to evaluate the control effect of endolysin LysCP28 on foodborne pathogens in food. To do that, we have purified the putative endolysin LysCP28 encoded by phage BG3P and determined its lytic activity against *C. perfringens* strains in vitro against both planktonic and sessile cells. Finally, the potential of endolysin LysCP28 as a biopreservative to reduce *C. perfringens* contamination in raw meat was also evaluated.

## 2. Materials and Methods

### 2.1. Bacterial Strains, Bacteriophages, and Growth Conditions

All of the test strains of *C. perfringens* mentioned in this work are listed in Appendix A. Seven reference strains and 89 strains, which were previously isolated at Binzhou, Rizhao, Xi’an, Mianyang, and Sihong by using the National Standard GB4789.13–2012 and Rood et al. method [23], were indicated. Anaerobic Meat Liver (AML) Broth (Qingdao Hope Bio-Technology Co., Ltd., Qingdao, China) was used for anaerobic culture of all strains at 37 °C. The strains were supplemented with 30% glycerol and stored in −80 °C Brain Heart Infusion (BHI) Broth (Qingdao Hope Bio-Technology Co., Ltd., Qingdao, China). CM0587 Perfringens Agar Base (OXOID Co., Ltd., Basingstoke, UK) was used for *C. perfringens* cell counting. TY medium (media component: 1 g yeast extract, 1 g sodium chloride, 1 g soy peptone, 3 g tryptose, and 0.6% agar every 200 mL) (OXOID Co., Ltd., Basingstoke, UK) was used for the double-layer assay. The bacteriophage vB_CpeS_BG3P was propagated and purified as previously described [22].

### 2.2. Cloning, Expression, and Purification of Endolysin LysCP28

The 10^9^ PFU/mL phage vB_CpeS_BG3P was used for whole-genome extraction via the phenol-chloroform-isoamyl alcohol method described by Zhou et al. [24]. The LysCP28 encoding gene (orf28) was amplified from the genomic DNA of the phage by polymerase chain reaction (PCR) using primers lys28F (5′-GCG GGA TCC ATG AAA ATA ATA CAA TCA AAT ATC CAT TTT-3′) and lys28R (5′-CGC AAG CTT TTA GTC TTT TTT AAT ATA TTT TGC GGA-3′). The gene was also optimized based on *E. coli* codon usage with the GenScript Rare Codon Analysis Tool. The optimized sequence was cloned into pET28a (Novagen, Madison, WI, USA), and plasmids with the correct insert were transformed into competent *E. coli* BL21(DE3). Expression of the recombinant LysCP28 was induced with 0.5 mM isopropyl β-D-1- thiogalactopyranoside (IPTG) when cultures reached an OD_600_ of 0.6–0.8, and then incubated at 30 °C for another 15 h. Bacterial cells were suspended in lysis buffer (50 mM Tris-HCl, 100 mM sodium chloride, pH 7.5) and disrupted with sonication (Branson Ultrasonics, Shanghai, China). The lysates were centrifuged at 15,000× *g* for 40 min at 4 °C. The supernatant was mixed with HisPur™ Ni-NTA Resin (Thermo Fisher Scientific, Waltham, MA, USA) and eluted according to the manufacturer’s instructions. Determination of the purified protein concentration was carried out by using a BCA Protein Assay kit (Shanghai Yase Biotechnology Co., Ltd., Shanghai, China).

### 2.3. Determination of the Activity Spectrum Using Plate Lysis Assay

To determine the lytic spectrum of endolysin LysCP28, a wide collection of *C. perfringens* strains and other gram-positive (*S. aureus*) and gram-negative (*E. coli* and *Salmonella*) bacteria were tested using a plate lysis assay. A plate lysis assay was conducted as previously described [25]. Briefly, 96 *C. perfringens* strains, including the reference strains and 89 isolates, were grown to mid-log phase (OD_600_ = 0.6). Each culture (0.5 mL) was mixed with 4 mL TY molten soft agar and then distributed evenly to solidify on a TY standard agar plate. Aliquots of purified LysCP28 (10 μL, 100 µg/mL) were spotted on the plates and incubated for 20 h at 37 °C. The diameter of the clear zone reflecting the inhibition of bacterial growth was recorded. According to the size of the lytic halo (large diameter ≥ 2 cm, ++, small 1 cm ≤ diameter < 2 cm, +, no lytic zone) the activity of LysCP28 on each bacterial strain was classified. Each experiment was repeated three times.

### 2.4. Time-Kill Assay and Stability of LysCP28

*C. perfringens* ATCC 13124 and *C. perfringens* J21 were grown to mid-log phase (OD_600_ = 0.4–0.6), and the cells were then pelleted (3000× *g*, 15 min, 4 °C), washed three times in sterile distilled water or PBS solution, and then suspended to an OD_600_ of 0.5–1.0. Purified LysCP28 was serially diluted in PBS solution (1:2) (20 μg/mL, 10 μg/mL, 5 μg/mL, 2.5 μg/mL) and mixed with fresh *C. perfringens* cultures (OD_600_ = 0.6, 2 × 10^7^ CFU/mL). Samples were incubated statically at 37 °C for a total of 6 h, and their OD_600_ was measured every 30 min. As a control, bacterial growth was also determined in the absence of endolysin for a period of 6 h. Each experiment was repeated three times.

For stability assays, LysCP28 (10 μg/mL) were placed in 40 mM (pH 7.2) boric acid–phosphoric acid (BP buffer, Shanghai Yuanye Biotechnology Co., Ltd., Shanghai, China) and pre-exposed to different temperatures (4 °C, 25 °C, 37 °C, 42 °C, 50 °C, and 60 °C) for 2 h. Bacterial cells (OD_600_ = 0.6), previously washed and suspended in BP buffer, were added to the pre-exposed protein LysCP28 at 37 °C for 2 h. The equal BP buffer was added to the same treatment of bacterial cells as the control samples. After that, the activity of the protein was measured as indicated above. The results were obtained as mean and standard deviation from triplicate measurements. For measurement of lytic activity at different pH levels, bacterial cells were washed and suspended in BP buffer (final concentration OD_600_ = 0.6) with a pH range from 4 to 11. Then, LysCP28 was added (final concentration 10 μg/mL) and incubated at 37 °C for 2 h. The control samples were placed in BP buffer via the same methods. The results were obtained as mean and standard deviation from triplicate measurements.

### 2.5. Observation of Cultures Via Scanning Electron Microscope (SEM)

*C. perfringens* J21 was grown to mid-log phase (OD_600_ = 0.4–0.6) and cells washed with PBS buffer and suspended to OD_600_ = 0.6. LysCP28 was added (final concentration 10 μg/mL) to the culture and incubated at 37 °C for 30 min. The bacterial mixtures were sampled at 0, 10, 20, and 30 min and centrifuged 3000× *g* for 5 min at 4 °C. The pellets were treated with precooled 2.5% glutaraldehyde solution and fixed at 4 °C for 2–12 h. The samples were then washed twice with 10 mM sodium phosphate (pH 7.5) and dehydrated in 50, 70, 90, and 99.5% ethanol. Subsequently, the ethanol was replaced with 50% ethanol−50% isoamyl acetate and 100% isoamyl acetate. The samples were dried using a critical-point dryer and subsequently sputter-coated with platinum using an ion sputtering machine. The grid was then loaded into a ZEISS EVO-LS10 scanning electron microscope (ZEISS, Germany), and the bacteria were examined at a 5000× magnification.

### 2.6. Biofilm Assays

Overnight cultures of *C. perfringens* were resuspended in AML broth at a density of 0.5 MacFarland supplemented with 10 mM filter-sterilized glucose. Biofilms were obtained as previously described [26]. Volumes of 200 μL of the cultures were added to 96-well polystyrene microplates (Costar^®^ #3595, Corning Incorporated, Corning, NY, USA) [27] and incubated anaerobically at 37 °C for 3 days in a sealed container. Then, planktonic cultures were removed, and the plate was washed twice with distilled water and left to dry for 24 h. The crystal violet assay was used to quantify biofilm formation. A volume of 200 μL of 0.1% crystal violet (Sigma Aldrich, Oakville, ON, Canada) was added to the wells and incubated for 3 min. The plate was washed twice, and 200 μL of 95% ethanol was added and then removed. Each strain was performed on 9 replicates with random selections from every source. The absorbance was measured at 570 nm. Based on the A_570_ values, biofilm formation was designated as strong (≥1.10, 1 × 10^9^ CFU/mL), moderate (0.70–1.09, 6 × 10^8^–1 × 10^9^ CFU/mL), weak (0.35–0.69, 2.5 × 10^8^–6 × 10^8^ CFU/mL), or absent (<0.35, 2.5 × 10^8^ CFU/mL) [28,29].

To test the efficacy of LysCP28 in *C. perfringens* biofilm removal, 20 μL of an overnight culture was added to 180 μL of AML broth in a polystyrene microplate. After 1 d, 2 d, and 3 d of incubation at 37 °C, the supernatants were discarded, and the wells were washed with PBS to remove unattached cells. PBS buffer (100 μL) was added to the control wells, whereas LysCP28 (100 μL, 0 μg/mL−300 μg/mL) was added to the treatment wells. After 24 h of incubation at 37 °C, the microplate was stained with crystal violet, and the results were recorded.

As described previously, several assays were performed on polystyrene microplates to evaluate the efficacy of LysCP28 in inhibiting biofilm formation [26]. Briefly, 10 μL of an overnight culture of *C. perfringens* was added to 90 μL of AML broth and poured in each well in a 96-well polystyrene microplate. In the control wells, 100 μL of phosphate buffer were added to the cell suspension. In the treatment wells, 100 μL of serially diluted (1:2) LysCP28 (300 μg/mL) was added, and the microplate was then incubated for 3 d at 37 °C. Total biomass was determined after biofilms were stained with crystal violet as indicated above.

### 2.7. Lytic Activity in Duck Meat

Duck breast meat was cut aseptically into 2 cm × 2 cm squares (about 1 g) and disinfected with UV to minimize contamination by spoilage microorganisms. *C. perfringens* ATCC 13124 was grown overnight to reach an OD_600_ of 1.0. For these assays, sterilized duck squares were immersed in a suspension containing *C. perfringens* overnight cultures diluted with PBS (1:1) for 15 min at an inoculum volume of approximately 1 × 10^3^–1 × 10^4^ CFU/g [30]. After that, the duck squares were removed and placed in 24-well plates (Costar^®^ #3595, Corning Incorporated, Corning, NY, USA). Finally, 0.1 mL of LysCP28 (5–500 µg/mL) was spotted onto the duck squares, and the same capacity of PBS was dropped onto inoculated duck squares as a negative control by 0.1 mL pipette accurately. Three replicates were prepared and measured for each treatment. The 24-well plates were stored immediately in a Ziploc bag for incubation at 4 °C for 3 d. The duck squares were transferred to test tubes at 0, 1, 12, 24, 48, and 72 h, 10 mL PBS solution was added, and the squares were shaken at room temperature for 30 min. The collected liquid portion was transferred to a sterile tube and centrifuged at 3000× *g* for 10 min. Cells were harvested and then enumerated on TSC and SFP media. The number of bacteria were estimated by the dilution gradient multiplied by the number of bacteria harvested.

### 2.8. Statistical Analysis

Statistical analyses were conducted on PRISM software (GraphPad PRISM v8.0, GraphPad Software, LLC, San Diego, CA, USA). Comparisons were analyzed using nonparametric one-way analysis of variance (ANOVA) with Bonferroni’s multiple-comparison post-hoc test.

## 3. Results

### 3.1. Identification and Overexpression of a New Endolysin (LysCP28)

The bioinformatic analysis of the genome sequence of phage vB_CpeS_BG3P allowed the identification of the putative endolysin gene (orf28, 1008 bp). The encoded protein, designated as LysCP28, was 335 amino acids long (38.8 kDa) and highly homologous to endolysins of the *Clostridium* phages phiSM101 (64.96%) and phiS63 (65.03%) (Appendix A). Pfam and Conserved Domain Database analysis showed a putative glycosyl hydrolase domain (glycosyl hydrolases family 25) or lysozyme at the N- terminal end of LysCP28 and a cell wall binding domain (CBD) formed by two adjacent SH3_3 domains at the C-terminus (Appendix A).

In order to further study this protein, a codon-optimized version of the orf28 sequence was used to produce a recombinant histidine-tagged LysCP28 in *E. coli* BL21(DE3). Overexpression of the protein and analysis of the purified fractions with SDS-PAGE revealed that LysCP28 was predominantly located in the soluble fraction and was easily purified to almost complete homogeneity using Ni-NTA resin under native conditions (Appendix A). Protein eluted with 100 mM imidazole resulted in high concentration (879.63 μg/mL) (Appendix A).

### 3.2. LysCP28 Kills a Wide Range of C. perfringens Strains

LysCP28 exhibited a broad spectrum of lytic activity, as most of the *C. perfringens* strains (77 of 96, or 80.2%) from different sources were lysed by LysCP28 (Appendix A). The diameter of the halo indicated different sensitivity ranges; 27 *C. perfringens* strains showed a strong lytic zone (≥2 cm), whereas a halo of about 1 to 2 cm was observed for 50 strains. Of note, LysCP28 had lytic activity against all *C. perfringens* reference strains and also against all tested type isolates (A, B, C, and D). No lytic zones were observed on *S. aureus*, *E. coli*, and *S.* Enteritidis strains.

### 3.3. The Lytic Activity of LysCP28 Was Confirmed Using C. perfringens Planktonic Cultures

The antimicrobial activity of LysCP28 against *C. perfringens* was determined with in vitro tests using a range of both protein concentrations and bacterial densities. Two different strains (ATCC 13124 and J21) were tested to compare the inhibitory effectiveness of endolysin on bacteria of different origins. The results indicated that LysCP28 was capable of inhibiting *C. perfringens* ATCC 13124 growth at all tested protein concentrations as compared with the control culture (Figure 1A). In addition, a fast decrease in optical density within 2 h was observed in all treated cultures, which then remained constant with a slight increase after 5 h. By adding 20 μg/mL, the lowest optical density (OD_600_ dropped to 0.09 from 0.61, 10^1^ CFU/mL from 10^7^ CFU/mL, *p* < 0.001) was reached at 1.5 h and remained even after 6 h (OD_600_ = 0.12, 10^1^ CFU/mL, *p* < 0.001). Interestingly, the lowest tested LysCP28 concentration (2.5 μg/mL) was enough to significantly reduce *C. perfringens* ATCC 13124 growth as compared with the control.

Similar to ATCC 13124, *C. perfringens* J21 turned out to be sensitive to endolysin and was strongly inhibited at all tested protein concentrations (Figure 1B). Thus, at 20 μg/mL, a dramatic reduction in optical density (from 1.0 to 0.1, from 10^1^ CFU/mL to 10^8^ CFU/mL, *p* < 0.001) within 30 min of incubation was observed. Still, the lowest protein concentration used (2.5 μg/mL) reduced optical density within 1.5 h and retained the inhibitory effect for at least 6 h (OD_600_ = 0.09, 10^1^ CFU/mL, *p* < 0.001). Overall, the inhibitory effect of LysCP28 was dose-dependent and more effective against strain J21 than against the reference strain ATCC 13124.

The microscopic effect of the endolysin activity on clostridia cells was observed via SEM on treated bacteria. The cultures of *C. perfringens* J21 without treatment were made up of cells with rod-shaped structures, a full appearance, and smooth cell surfaces without defects (Figure 2A). However, when LysCP28 was applied for 10 min before microscopic observation, bacterial cells showed surface defects, and they were attached to each other (Figure 2B). Furthermore, almost all *C. perfringens* cells were completely lysed after 20 min of exposure to LysCP28 (Figure 2C), and only cell debris was observed after 30 min (Figure 2D).

### 3.4. Tolerance of LysCP28 to Stress Conditions

The use of endolysins as antimicrobials requires stable activity in a wide range of conditions, including temperatures and pH levels. A very similar pattern of lytic activity was observed for both strains (*C. perfringens* ATCC 13124 and J21), as the optimal activity (the highest OD_600_ decrease) was obtained at 42 °C (Figure 3A). At this temperature, *C. perfringens* strain J21 turned out to be more sensitive than ATCC 13124. As was expected, temperatures of 25 °C and 37 °C were also suitable for protein activity. Over 50 °C, the inhibition in cultures growth was reduced but still significant as compared with the control.

Regarding pH sensitivity, LysCP28 was active in the range of pH 5 to 9 against both tested strains. Again, the *C. perfringens* J21 strain had higher sensitivity to endolysin in most of the tested conditions (Figure 3B). Comparing with the control, the optimal activity was obtained at pH 7, although no statistically significant differences were observed at pH 8. Overall, at acidic pH levels, the activity of LysCP28 was higher than at basic pH levels.

### 3.5. The Antibiofilm Activity of LysCP28

Due to some *C. perfringens* strains’ ability to form biofilms, we wanted to examine the effect of endolysin to inhibit their biofilm-forming ability and to remove preformed biofilms. For this, it was essential to ensure that the selected strains represent different degrees of biomass attachment. To find out about this, strains from different origins (*n* = 12) were selected, and biofilms were obtained after 3 days of incubation at 37 °C. The total biomass was determined after crystal violet staining. Most of *C. perfringens* strains were able to form biofilms, and the optical density values (A_570_) of total biomass ranged from 0.19 to 1.13 (Figure 4).

Among the studied strains, *C. perfringens* J21 was found to be a strong biofilm producer with OD values ranging from 0.67 to 1.13 (*p* < 0.001). *C. perfringens* ATCC 13124 had a moderate ability to form biofilms with OD values ranging from 0.23 to 0.65 (*p* < 0.01). The rest of the strains (*n* = 10) were weak biofilm producers (*p* < 0.05) (Figure 4). Therefore, we selected *C. perfringens* J21 for biofilm inhibition assays using endolysin. 

Three-day-old biofilms formed by this strain were treated with endolysin LysCP28 (final concentration from 0 to 300 µg/mL) and incubated for 24 h at 37 °C. When a low dose of (18.75 µg/mL) LysCP28 was added to the biofilms, the biomass value dropped from 0.79 (control samples) to 0.15, which is more than five times lower than the control (0 μg/mL) (*p* < 0.001) (Figure 5A). However, when 9.37 µg/mL (or less) LysCP28 was added to *C. perfringens* J21 biofilms, the OD value was not significantly reduced (*p* < 0.05) (Figure 5A). Thus, 18.75 µg/mL was the minimum concentration to remove biofilms formed by this strain. Protein concentrations above that level did not result in higher efficiency for biomass removal.

Additionally, LysCP28 (0–300 µg/mL) was added to biofilms of different maturation times (incubated a 37 °C for 3 days). When LysCP28 was added to biofilms formed by strain J21, A_570_ significantly decreased as compared with untreated samples. Thus, the biomass measure in control samples was about 0.91, whereas 1 d-old biofilms treated with endolysin decreased absorbance to 0.24 and 0.29 for those 2 days old, and to 0.21 for 3-day-old biofilms (day 1 and day 3 *** *p* < 0.001, day 2 ** *p* < 0.01) (Figure 5B). Therefore, LysCP28 displayed strong antibiofilm effect against preformed *C. perfringens* biofilms in both the early and late stages of biofilm maturation.

### 3.6. Preservative Potential in Raw Duck Meat

*C. perfringens* is one of the major foodborne pathogens closely related to poultry meat contamination. In this context, the antimicrobial activity of endolysin LysCP28 to inhibit the growth of this pathogen was tested in duck meat. To do that, meat squares were contaminated with *C. perfringens* ATCC 13124 (2.3 × 10^3^ CFU/g), treated with endolysin (in concentrations from 5 µg/mL to 500 µg/mL), and stored at 4 °C. Overall, the difference was statistically significant in terms of viable cells (Figure 6) between the endolysin treatment and control samples. Endolysin LysCP28 typically showed a dose-dependent inhibition of *C. perfringens*. Thus, high doses of LysCP28 (500 µg/mL) were able to completely eliminate the pathogen after a thorough 24 h treatment. Bacterial concentration was reduced by approximately 3.2-log and 3.08-log with 100 µg/mL and 50 µg/mL LysCP28, respectively, after 48 h incubation. As expected, the inhibition efficiency decreased as the endolysin dose decreased. A low dose of LysCP28 (5 µg/mL) was not as effective at removing *C. perfringens* as the high dose, and an approximately 1.89-log reduction of *C. perfringens* was still observed after 72 h of treatment. After 72 h incubation, the density of *C. perfringens* was still close to the initial concentration (>90%) in the control samples without endolysin.

## 4. Discussion

The rapid emergence of resistant bacteria worldwide has imperiled the efficacy of antibiotics, resulting in drastic measures such as the usage of these compounds as growth promoters and as therapeutics for livestock and poultry. Therefore, there is a need for the development of alternatives in order to control some diseases, such as those caused by *C. perfringens*. This bacterium has a strong ability to acquire antibiotic resistances [31], produce spores, and form biofilms, which contributes to its persistence in the environment. Indeed, poultry is frequently colonized and infected by *C. perfringens* type A producing the alpha toxin, which causes necrotic enteritis, and has resulted in a reduction in the productivity of the poultry industry [32]. In addition, this pathogen has been detected in meat [33]. In this regard, some alternatives to reduce this pathogen in the food chain have been explored. These alternatives include improving animal health, such as by using vaccines [32] natural antimicrobials [34], enzymes [35], and probiotics [36]. All of these strategies are intended to reduce the presence of the pathogen in the production chain, but additional preservation techniques also have to be applied to the meat, a minimally processed product with risk of contamination by spore-forming bacteria [37].

In this context, we explored the use of endolysins, which have been previously proposed as antimicrobials for application in food [38]. A putative endolysin-encoding gene was identified from the phage vB_CpeS_BG3P genome [22]. The sequence analysis of the protein (LysCP28) revealed that the N-terminal catalytic domain would have glycosyl hydrolase activity, and the C-terminal end includes two SH3_3 domains. The amino acid sequence was highly homologous to the encoded endolysin (Psm) by episomal phage phiSM101, for which the mode of action has been determined [39]. Endolysins with glycosyl hydrolase activity have been proposed as suitable for oral administration to animals because they have showed good activity against strains isolated from this source [40]. In fact, overlooking the use of endolysins for this purpose, the phage endolysin PlyCP41 was expressed in *Nicotiana benthamiana* plants that are suitable to be included in animal feeds [41].

In the case of LysCP28, it showed an extensive spectrum of lytic activity against the tested *C. perfringens* strains in contrast to the φ3626 phage endolysin Ply3626, which only had activity on about 22% of the *C. perfringens* strains [17]. For most spore-forming bacteria, the ability of endolysins to lyse spores could not be demonstrated. However, recently [42] described the endolysin LysPBC2 from *Bacillus cereus* phage PBC2, which exhibited much extensive lytic activity against all *Listeria*, *Bacillus*, and *Clostridium* species, and which contains a spore binding domain (SBD) that binds to *B. cereus* spores but not to vegetative cells specifically.

In addition to the lytic spectrum, it is also important to maintain protein stability, mainly when the final application is healing chickens with gastrointestinal tract infections, which have a rectal temperature between 40.6 and 43.0 °C. LysCP28 has optimal activity between 37 °C and 42 °C, which shows the endolysin LysCP28 would be suitable for this application. In some cases, the stability of the endolysin can be improved by fusion of the protein to a catalytic domain of the thermophilic *Geobacillus* endolysin PlyGVE2, which increased the stability from 42 °C up to 55 °C [43]. Moreover, some endolysins, such as LysCPS2, were confirmed to be highly thermally stable, maintaining 30% of their lytic activity after incubation at 95 °C for 10 min [19]. Therefore, this lysase can be modified according to different uses to make it more valuable. In addition, we were interested to know the activity at different pHs, especially the meat pH (5 to 6), at which the protein retained good lytic activity.

For biofilm-forming bacteria, the lytic activity of endolysins must also be assessed, not only against planktonic bacteria. In this study, the *C. perfringens* J21 strain showed higher capacity for biofilm formation than other isolates and was selected to check biofilms’ sensitivity to LysCP28. We found that this protein not only prevented the formation of biofilm, but also disrupted biofilms that had already formed. This result was not totally new, as other endolysins were previously demonstrated to be active against biofilms [44,45], but to our knowledge, this is the first time this has been proved for *C. perfringens* biofilms. Moreover, our study also indicated that a low dose of endolysin did not induce biofilm formation such as it happens when using antibiotics [46]. Actually, for some endolysins, it has been demonstrated that they do not induce biofilm formation [45]; rather, they inhibit the formation of this structure [47], which constitutes an advantage over antibiotics. Although low-dose endolysin has limited inhibitory ability on biofilms, low doses of endolysin will degrade over time, making them unlikely to induce biofilm growth.

Contaminated raw poultry products are recognized as the main source of *C. perfringens* infections, which is a potential threat to public health [48,49]. In recent years, 207 samples (48.82%) of *C. perfringens* were isolated from four duck farms in Shandong Province [50]. Liu et al. [51] tracked *C. perfringens* from a commercial duck breeding enterprise in Shandong Province along the production chain, which proved the possibility of chain propagation. 

Previous studies have pointed the usefulness of endolysins for controlling *Clostridium* contamination in food. Mayer et al. [52] used the endolysin CTP1L against *C. tyrobutyricum* cells in milk and even in cheese manufacturing processes by the use of *Lactococcus lactis* endolysin-producing strains [53]. Cho et al. [21] added endolysin (10.2 µg/mL) LysCPAS15 to milk to inhibit *C. perfringens* by 3-log within 2 h at 37 °C. Kazanavičiūtė et al. [54] showed that the inhibitory effect of nisin on *C. perfringens* was lower than plant-expressed lysins (ZP173 and CP25L) in cooked meat. In addition, endolysin CP25L was highly efficient in turkey meat, and bacterial titers were decreased by 4-log CFU compared to a control even after 43 h of incubation. In this study, 500 µg/mL endolysin LysCP28 reduced *C. perfringens* by more than 3.4-log CFUs compared with control samples in duck meat after 72 h. Although LysCP28’ bacteriolytic activity was slightly dose-dependent, it could also significantly reduce bacteria after 72 h in duck meat. From above, the phage endolysin LysCP28 was evaluated under conditions simulating a realistic model of food processing, preparation, preparation, and consumption, thus manifesting promise as a potential antibacterial agent to control *C. perfringens* in food.

## 5. Conclusions

Phage endolysin LysCP28 is an alternative to control *C. perfringens* biofilm formation and food contamination. LysCP28 prevented biofilm development and removed biofilms that had already formed. Based on this study, the potential application of phage-derived endolysins in the prevention and elimination of biofilm-related contamination in food processing is a promising biocontrol option. Moreover, LysCP28 showed high inhibition efficacy against *C. perfringens* in duck meat. Therefore, phage endolysins comply with Generally Recognized as Safe (GRAS) criteria and could be potentially used as a natural biocontrol preservative to reduce *C. perfringens* contamination during the processing stages of food production, which could effectively reduce the risk of *C. perfringens* contamination and the occurrence of poisoning events caused by *C. perfringens*.

## Figures and Tables

**Figure 1 foods-12-00411-f001:**
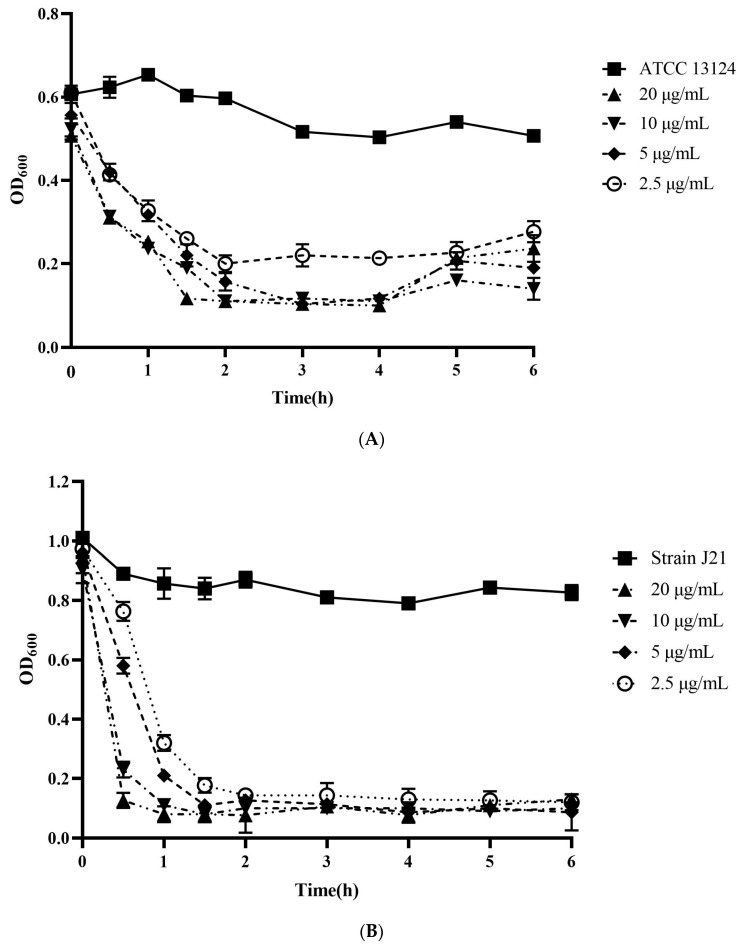
Time-kill assay of *C. perfringens* ATCC 13124 (**A**) and J21 (**B**) cultures treated with purified LysCP28. Optical density was measured periodically. *C. perfringens* ATCC 13124 and J21 without treatment were used as controls. Each test was assayed in triplicate.

**Figure 2 foods-12-00411-f002:**
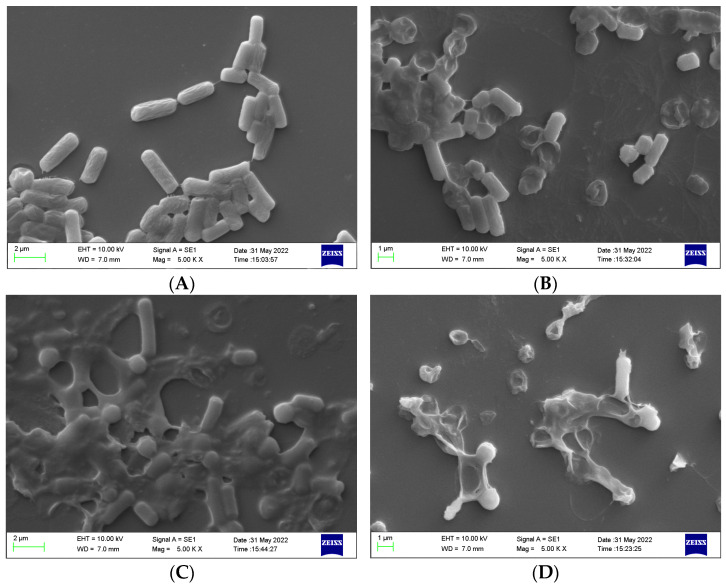
Microscopic images of *C. perfringens* cultures treated with LysCP28 (10 μg/mL) at 37 °C. Samples were taken at 0 min (**A**), 10 min (**B**), 20 min (**C**), and 30 min (**D**) and analyzed via scanning electron microscope (SEM).

**Figure 3 foods-12-00411-f003:**
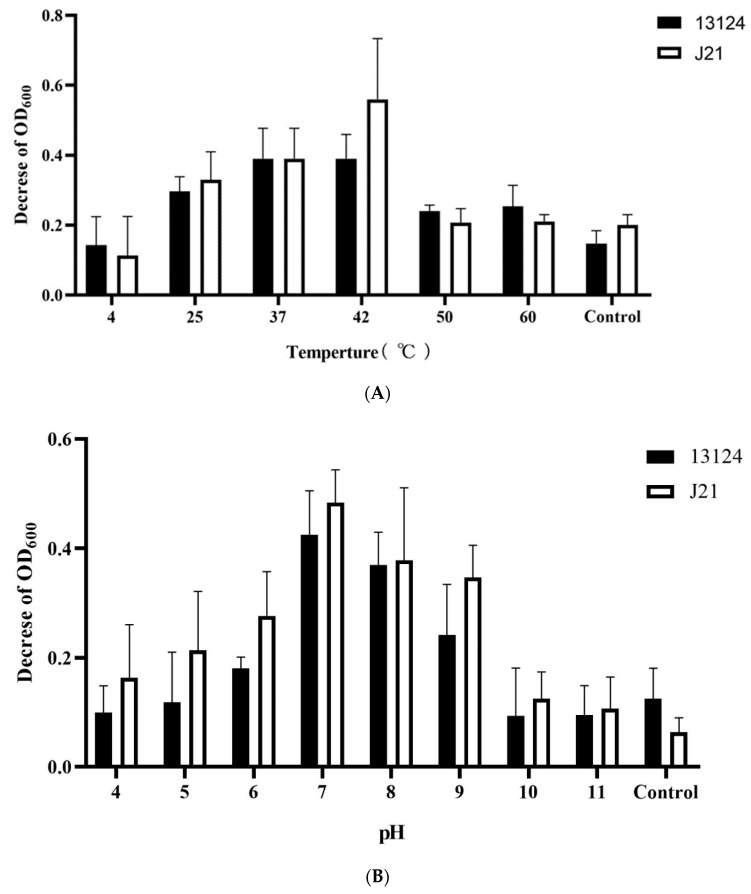
Lytic activity of endolysin LysCP28 in stress conditions. (**A**) The tolerance to temperature was measured after previous incubation for 2 h in a range of temperatures. The graph represents the difference between initial and final optical density of cultures treated with temperature-treated endolysin. (**B**) The activity in different pH levels was determined by analyzing the incubation of LysCP28 and cells over 2 h. The graph represents the difference between the initial and final optical densities of cultures treated with endolysin at different pH levels. OD_600_ = 0.2 (ATCC 13124, J21 = 2 × 10^2^ CFU/mL), OD_600_ = 0.4 (ATCC 13124 = 5 × 10^4^ CFU/mL, J21 = 1 × 10^5^ CFU/mL), OD_600_ = 0.6 (ATCC 13124, J21 = 2 × 10^7^ CFU/mL), and OD_600_ = 0.8 (ATCC 13124 = 4 × 10^7^ CFU/mL, J21 = 6 × 10^7^ CFU/mL).

**Figure 4 foods-12-00411-f004:**
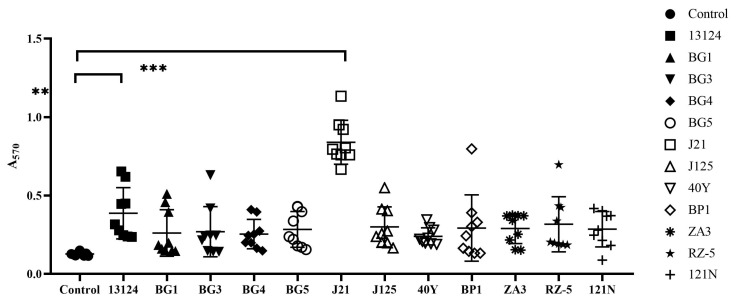
Biofilm formation by *C. perfringens* isolates after incubation at 37 °C for 3 days. Strains were selected from different origins. Each dot represents nine replicates for each strain. The bars represent the mean values of each group. The *p* values were calculated using a linear model for unequal variances. The A_570_ comparison between different groups was performed using an independent two-sample *t*-test. Asterisks represent significant differences between different groups (** *p* < 0.01, *** *p* < 0.001).

**Figure 5 foods-12-00411-f005:**
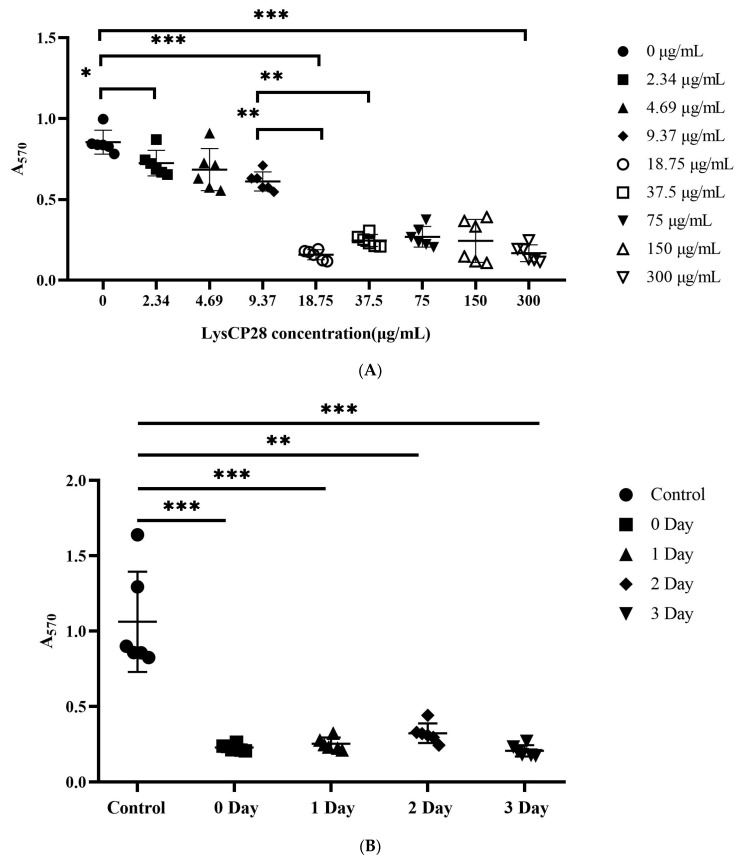
Efficacy of endolysin as antibiofilm agent against *C. perfringens* J21 strain. (**A**) Preformed biofilms (3-days-old) treated with different concentrations of LysCP28 over 24 h at 37 °C. (**B**) Inhibition in the biofilm formation (1-day-, 2-day, and 3-day-old) using endolysin LysCP28. The bars show the mean A_570_ ± SD and 95% confidence interval from six independent experiments. The bars without the letter A were significantly different from the control group. The A_570_ comparison between different groups was performed using an independent two-sample *t*-test. Asterisks represent significant differences between different groups (* *p* < 0.50, ** *p* < 0.01, *** *p* < 0.001).

**Figure 6 foods-12-00411-f006:**
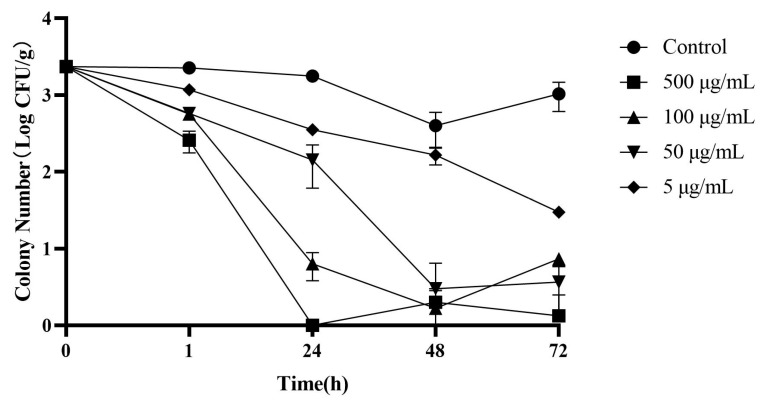
Inactivation of *C. perfringens* in duck meat by endolysin LysCP28 after incubation for 72 h at 4 °C. Changes of *C. perfringens* viable counts (log CFU/g) after treatment: 500 µg/mL (
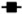
), 100 µg/mL (
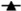
), 50 µg/mL (
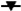
), and 5 µg/mL (
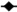
) were recorded. Control samples of 10^3^ CFU/g (
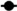
). Values correspond to the mean for three independent experiments.

## Data Availability

The data presented in this study are available on request from the corresponding authors.

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
