# Peer review of "A Broad-Spectrum Phage Endolysin (LysCP28) Able to Remove Biofilms and Inactivate Clostridium perfringens Strains"

_foods, 2023, doi:10.3390/foods12020411_

Round 1

Reviewer 1 Report

The paper structure and motivation are fine. The paper needs minor improvements, especially the results and discussion sections.

Major:

(1)    The methods used OD and Crystal violet are poor accurate. The determination of CFUs and molecular biology techniques must be included.

(2)    The discussion section must be improved. The sentences are poorly written and confusing.

Minor:

Figure 1 should be relocated to be first mention on the text.

Figure 2A. The temperature of the control must be specified. The information must be add to the subtitle.

Line 253: The authors must execute statistically analysis to state that the results are significatively lower.

Line 277: Remove “good”

Figure 5A. Is this figure correct? The information on the X axis is the same as the legend. The graphic must be improved. It is very difficult to analyse it.

Line 309: Rephrase the sentence

Author Response

Response to Reviewer #1

The paper structure and motivation are fine. The paper needs minor improvements, especially the results and discussion sections.

Major:

  1. The methods used OD and Crystal violet are poor accurate. The determination of CFUs and molecular biology techniques must be included.

Answer: Thank you for your suggestions. We used OD as a general method for biofilm determination, which can well illustrate bacterial concentration. Now CFU has been added in the article, and we cited related research, reference 26-28.

  • “Stepanovic, S., Vukovic, D., Dakic, I., Savic, B., Svabic-Vlahovic, M. A modified microtiter-plate test for quantification of staphylococcal biofilm formation. J Microbiol Methods. 2000, 40, 175–179”.
  • “Charlebois, A., Jacques, M., and Archambault, M. Biofilm formation of Clostridium perfringens and its exposure to low-dose antimicrobials. Front Microbiol. 2014, 5, 183”.
  • “García-Heredia, A., García, S., Merino-Mascorro, J. Á., Feng, P., Heredia, N. Natural plant products inhibits growth and alters the swarming motility, biofilm formation, and expression of virulence genes in enteroaggregative and enterohemorrhagic Escherichia coli. Food microbiol. 2016, 59, 124–132”.
  1. The discussion section must be improved. The sentences are poorly written and confusing.

Answer: Thank you for the suggestions. The discussion has been revised in manuscript.

(1) Line 354-356, we have revised to “The rapid emergence of resistant bacteria worldwide imperiling the efficacy of antibiotics, has resulted in a drastic control with the usage of these compounds as growth promoters and as therapeutics for livestock and poultry.”

(2) Line 396-398, we have revised to “Even, some endolysin such as LysCPS2 was confirmed to be highly thermally stable, maintaining 30% of its lytic activity after incubation at 95 °C for 10 min.”

(3) Line 416-421, we have revised to “Contaminated raw poultry products are recognized as the main source of C. perfringens infections, which is a potential threat to public health [48, 49]. In recent years, 207 samples (48.82%) of C. perfringens were isolated from 4 duck farms in Shandong Province [50]. Liu et al. [51] tracked C. perfringens from a commercial duck breeding enterprise in Shandong Province along the production chain, which proved the possibility of chain propagation.”

(4) Line 425-437, we have revised to “Cho et al. [21] added endolysin (10.2 µg/mL) LysCPAS15 in milk to inhibit C. perfringens by 3-log within 2 h at 37°C. Kazanavičiūtė et al. [54] showed that the inhibitory effect of nisin on C. perfringens was lower than plant-expressed lysins (ZP173 and CP25L) in cooked meat. In addition, endolysin CP25L was high-efficient in turkey meat, and bacterial titers were decreased by 4-log CFU compared to a control even after 43 h of incubation. In this study, 500 µg/mL endolysin LysCP28 reduced C. perfringens more than 3.4-log CFUs compared with control sample in duck meat after 72 h. Although LysCP28 performed a slightly dose-dependent bacteriolytic activity, it can also significantly decline bacteria after 72 h in duck meat. From above, phage endolysin LysCP28 was evaluated under conditions simulating realistically model food processing, preparation, preparation, and consumption, manifesting promise as a potential antibacterial agent to control C. perfringens in food.”

Minor:

  1. Figure 1 should be relocated to be first mention on the text.

Answer: We named according to the magazine format requirements. In line 241 - 251, there were supplementary figure 1S and figure 2S before figure 1 in manuscript. So, we named the figure 1.

  1. Figure 2A. The temperature of the control must be specified. The information must be add to the subtitle.

Answer: Line 642-644, the temperature of the control was incubated at 37℃, and all bacterium were incubated at 37℃ in figure 2A. The information has been described in the manuscript “2.5. Observation of cultures by scanning electron microscope (SEM)”.

  1. Line 253: The authors must execute statistically analysis to state that the results are significatively lower.

Answer: Line 269-270, the statistically analysis had been added in manuscript with “(OD600 dropped to 0.09 from 0.61, 101 CFU/mL from 107 CFU/mL, P<0.001)”.

  1. Line 277: Remove “good”

Answer: Line 290, we have removed in the manuscript.

  1. Figure 5A. Is this figure correct? The information on the X axis is the same as the legend. The graphic must be improved. It is very difficult to analyse it.

Answer: Line 703, we have modified figure 5A in manuscript, the X axis represents the LysCP28 concentration (μg/mL) with the range from 0-300 μg/mL.

  1. Line 309: Rephrase the sentence

Answer: Line 310-312, the sentence has been rephrased to “Most of C. perfringens strains were able to form biofilms, and the optical density values (A570) of total biomass ranged from 0.19 to 1.13” in manuscript.

Reviewer 2 Report

In the current study the author has expressed an endolysin (LysCP28) from vB_CpeS_BG3P phage of Clostridium perfringens. The protein was expressed in E. coli, nicely purified and tested for their potential to inactivate biofilm formed by C. perfringens. The author further tested the efficacy of LysCP28 to prevent C. perfringens growth in the duck meat at 4ºC. This is an interesting study that shows the potential of LysCP28 as a food biopreservative  for  reducing C. perfringens contamination during food processing. The manuscript is nicely written and data presented here are well organized. However, there are few major correction required to further improve the manuscript. I recommend to resubmit the manuscript after addition of these information. Here are my comments;

Comment1: The author has tested the enzyme for its growth reduction assay against two strains of C. perfringens which is not enough. The author needs to test few more strains of C. perfringens that were used in plate lysis assays to further strengthen the claim.

Comment 2: The author has performed the assays at 37˚C. Is this the Optimum Temperature for assay?  Why the assays at 42˚C was not performed as its shows higher activity when pre-incubated at 42˚C.

The author need to include the data of enzyme assays at different temperature i.e 25-50˚C.

Comment 3: Figure 3. It will be more appropriate if the reduction in bacterial cells are mentioned in CFU rather than decrease in OD.

Comment 4: Lane 332: The author stated thatendolysin decreased the absorbance to 0.24 and to 0.29 for those 2 d-old, and to 0.21  for 3 d-old biofilms”. How this decrease in absorbance can be explained quantitatively?

Comment 5: Lane 191: please include plates provider or brand

Comment 6: The endolysin LysCP28 is globular or modular in nature?

ForReference: https://www.sciencedirect.com/science/article/pii/S0141022921001046

Comment 7: There are a lot of grammatical mistakes in manuscript. The authors need to pay a careful attention to omit these errors

Comment 8: Overall the manuscript is well written and data is well presented, however, it will be more appropriate if the author provide future perspective of this study in conclusion.

Author Response

Response to Reviewer #2

In the current study the author has expressed an endolysin (LysCP28) from vB_CpeS_BG3P phage of Clostridium perfringens. The protein was expressed in E. coli, nicely purified and tested for their potential to inactivate biofilm formed by C. perfringens. The author further tested the efficacy of LysCP28 to prevent C. perfringens growth in the duck meat at 4ºC. This is an interesting study that shows the potential of LysCP28 as a food biopreservative for reducing C. perfringens contamination during food processing. The manuscript is nicely written and data presented here are well organized. However, there are few major correction required to further improve the manuscript. I recommend to resubmit the manuscript after addition of these information. Here are my comments;

Comment 1: The author has tested the enzyme for its growth reduction assay against two strains of C. perfringens which is not enough. The author needs to test few more strains of C. perfringens that were used in plate lysis assays to further strengthen the claim.

Answer: Thank you for your suggestion. In order to show the effect of the lysin in the submission, we only used two strains to describe the antimicrobial activity. ATCC 13124 represents the reference strain, whether strain J21 represents isolates from chicken. We also tested the effect with other strains, and the result was similar to J21 and ATCC 13124, so only two representative strains were put to show the antibacterial activity of the lysin.

Comment 2: The author has performed the assays at 37˚C. Is this the Optimum Temperature for assay? Why the assays at 42˚C was not performed as its shows higher activity when pre-incubated at 42˚C.

The author need to include the data of enzyme assays at different temperature i.e 25-50˚C.

Answer: Thank you for your suggestion. C. perfringens is an anaerobic bacterium with complex growth conditions. We tested the optimal activity of the lysin, and there was no significant difference between 37 ˚C and 42 ˚C, so we chose the more common 37 ˚C. We also tested the suitable temperature for ATCC 13124. It showed similar effect as the isolate J21. So we chose 37˚C for this test. We also referenced our previous research of the references 22 & 25.

  • “Huang, S., Tian, Y., Wang, Y., García, P., Liu, B., Lu, R., Wu, L., Bao, H., Pang, M., Zhou, Y., Wang, R., Zhang, H. The Broad Host Range Phage vB_CpeS_BG3P Is Able to Inhibit Clostridium perfringens Viruses. 2022, 14, 676”
  • “Tian, Y., Wu, L., Lu, R., Bao, H., Zhou, Y., Pang, M., Brown, J., Wang, J., Wang, R., Zhang, H. Virulent phage vB_CpeP_HN02 inhibits Clostridium perfringens on the surface of the chicken meat. Int J Food Microbiol. 2022, 363, 109514”

We selected 37 ˚C as the optimum temperature for assay. And the data of lysine assays at different temperature is in “3.4. Tolerance of LysCP28 to stress conditions’’ including different temperature from 4-60˚C in manuscript figure 3A. It also included 25 ˚C- 50˚C.

Comment 3: Figure 3. It will be more appropriate if the reduction in bacterial cells are mentioned in CFU rather than decrease in OD.

Answer: Thank you for your suggestion. The purpose of this experiment was to demonstrate the tolerance of LysCP28, so we used OD to illustrate. And We also added the CFU to the manuscript in line 651-653, “OD600=0.2 (ATCC 13124, J21=2×102 CFU/mL), OD600=0.4 (ATCC 13124=5×104 CFU/mL, J21=1×105 CFU/mL), OD600=0.6 (ATCC 13124, J21=2×107 CFU/mL), and OD600=0.8 (ATCC 13124=4×107 CFU/mL, J21=6×107 CFU/mL).”

Comment 4: Lane 332: The author stated that “endolysin decreased the absorbance to 0.24 and to 0.29 for those 2 d-old, and to 0.21 for 3 d-old biofilms”. How this decrease in absorbance can be explained quantitatively?

Answer: We have explained in the method that the biofilm remove test is carried out by crystal violet method. The absorbance value represents the change of the biofilm. Therefore, the value detected by crystal violet staining showed the function of the endolysin for removing biofilm. This is a common method of biofilm, which can show the remove effect of biofilm.

Comment 5: Lane 191: please include plates provider or brand

Answer: Line 219-220, we have added the provider “(Costar ® #3595, Corning Incorporated, Corning, NY, USA)” in manuscript.

Comment 6: The endolysin LysCP28 is globular or modular in nature?

For Reference:https://www.sciencedirect.com/science/article/pii/S0141022921001046

Answer: According to the relevant references and the comparative analysis of the relevant sequences, LysCP28 may be modular in nature, but in this study we has not carried out in-depth analysis on this aspect, focusing on the role of protein. Thank you very much for your suggestions. We will conduct in-depth analysis of its structure in the follow-up study.

 “Hermoso, J. A., Monterroso, B., Albert, A., Galán, B., Ahrazem, O., García, P., Martínez-Ripoll, M., García, J. L., & Menéndez, M. Structural basis for selective recognition of pneumococcal cell wall by modular endolysin from phage Cp-1. Structure (London, England: 1993). 2003, 11, 1239–1249.”

Comment 7: There are a lot of grammatical mistakes in manuscript. The authors need to pay a careful attention to omit these errors

Answer: Thank you for your suggestion. We have revised carefully in manuscript.

Comment 8: Overall the manuscript is well written and data is well presented, however, it will be more appropriate if the author provide future perspective of this study in conclusion.

Answer: Thank you for your suggestion. Line 438-449, the conclusion has been added in manuscript.

Reviewer 3 Report

Manuscript Number: foods-2108077

Title:  A broad-spectrum phage endolysin (LysCP28) able to remove biofilms and inactivate Clostridium perfringens strains

The submitted manuscript presents results of the study on the lytic activity of an endolysin LysCP28 encoded by orf28 from C. perfringens bacteriophage BG3P against planktonic and sessile cells of C. perfringens strains. In addition, the authors evaluated the bio-preservative potential of the aforementioned endolysin to reduce C. perfringens contamination in real duck meat samples. In my opinion, the study presents an interesting approach to the problem under consideration and thus it constitutes valuable extension of the knowledge related to antimicrobial and antibiofilm bio-agents. Besides the scientific character, the presented research has also an application nature. C. perfringens strains exhibit persistence to environment factors due to their strong ability to acquire antibiotic resistances, spore production and biofilm formation. The results of the study indicate that the lytic activity of tested endolysin LysCP28 can be used to reduce these strains resistance and improve the health of animals and eliminate the presence of the C. perfringens strains in the food production chain.

The authors correctly defined the research goal and supported its purposefulness with the analysis of the literature. The results are also clearly presented and discussed with previous research. The article is fairly written and organized. Below, there are minor suggestions that could further improve the manuscript:

Figure 1. I suggest adding the word "strain" to the species numbers in the legends.

Line 329: It should be “37ºC” instead of 37oC”

Line 330: It should be “A570” instead of A570”

Lines 3 and 334: The bacterial specie should be written in italics.

Line 169: Shouldn’t it be “(≥ 1.10)” instead of “(> 1.10)”

Line 191: By mistake, the authors left redundant text of the note “(please include plates provider or brand)”.

Author Response

Response to Reviewer #3

Title:  A broad-spectrum phage endolysin (LysCP28) able to remove biofilms and inactivate Clostridium perfringens strains

The submitted manuscript presents results of the study on the lytic activity of an endolysin LysCP28 encoded by orf28 from C. perfringens bacteriophage BG3P against planktonic and sessile cells of C. perfringens strains. In addition, the authors evaluated the bio-preservative potential of the a forementioned endolysin to reduce C. perfringens contamination in real duck meat samples. In my opinion, the study presents an interesting approach to the problem under consideration and thus it constitutes valuable extension of the knowledge related to antimicrobial and antibiofilm bio-agents. Besides the scientific character, the presented research has also an application nature. C. perfringens strains exhibit persistence to environment factors due to their strong ability to acquire antibiotic resistances, spore production and biofilm formation. The results of the study indicate that the lytic activity of tested endolysin LysCP28 can be used to reduce these strains resistance and improve the health of animals and eliminate the presence of the C. perfringens strains in the food production chain.

The authors correctly defined the research goal and supported its purposefulness with the analysis of the literature. The results are also clearly presented and discussed with previous research. The article is fairly written and organized. Below, there are minor suggestions that could further improve the manuscript:

  1. Figure 1. I suggest adding the word "strain" to the species numbers in the legends.

Answer: Line 674, we have added the word "strain" to the species numbers in the legends (figure 1) in manuscript.

  1. Line 329: It should be “37ºC” instead of “37oC”

Answer: Line 329, we have revised in manuscript.

  1. Line 330: It should be “A570” instead of “A570”

Answer: Line 330, we have revised in manuscript.

  1. Lines 3 and 334: The bacterial specie should be written in italics.

Answer: Line 2 and 335, we have revised in manuscript.

  1. Line 169: Shouldn’t it be “(≥ 1.10)” instead of “(> 1.10)”

Answer: Line 194, we have revised to “(≥ 1.10)” in manuscript.

  1. Line 191: By mistake, the authors left redundant text of the note “(please include plates provider or brand)”.

Answer: Line 219-220, we have revised the provider “(Costar ® #3595, Corning Incorporated, Corning, NY, USA)” in manuscript.

Reviewer 4 Report

 A broad-spectrum phage endolysin (LysCP28) able to remove biofilms and inactivate Clostridium perfringens strains

The abstracts need to be revised for clarity. Some sentences are grammatically not correct.

Line 189: why the meat pieces were socked for 15 minutes? A simple 60 to 120 second dip would be enough for the organisms to stick over the surface of meat.

Line 191: replace the comments with the company name.

Line 191-192: how the 0.1ml of LysCP28 were spotted over the surface of the meat?

Lien 196: Why 1:5 dilution was made instead of 1:10?? Ad how the number of bacteria were estimated then?

Line 223: 2.3: please take the methodology part of this section to the methodology part. Don’t put the methods in the results part.

Some scientific name of not italic.

Some experiments are discussed in the results but missing in the methodology part.

Please read the manuscript thoroughly and correct write the missing methods.

Methods detail of 3.1 of the results section is missing in methodology.

Methodology of 3.2, 3,4 is missing  

There are some formatting mistakes in the text of the manuscript body

Conclusion is missing??

Author Response

Response to Reviewer #4

 A broad-spectrum phage endolysin (LysCP28) able to remove biofilms and inactivate Clostridium perfringens strains

  1. The abstracts need to be revised for clarity. Some sentences are grammatically not correct.

Answer: Thank you for your suggestions. We have revised carefully in manuscript.

  1. Line 189: why the meat pieces were socked for 15 minutes? A simple 60 to 120 second dip would be enough for the organisms to stick over the surface of meat.

Answer: Line 217, C. perfringens is gram-positive anaerobe with complex growth conditions, we have tested that 15 minutes may let the strains adsorbed much better  on the surface of the meat pieces. Moreover, we also reference other methods. We found the Gram positive Listeria monocytogenes also need 15 minutes to adsorb on the surface of meat. We cited the related research reference 29 “Soni, K. A., Nannapaneni, R., Hagens, S. Reduction of Listeria monocytogenes on the Surface of Fresh Channel Catfish Fillets by Bacteriophage Listex P100. Foodborne Pathog Dis. 2010, 7, 427–434.”

  1. Line 191: replace the comments with the company name.

Answer: Line 219-220, we have replaced to “(Costar ® #3595, Corning Incorporated, Corning, NY, USA)” in manuscript.

  1. Line 191-192: how the 0.1ml of LysCP28 were spotted over the surface of the meat?

Answer: Line 222, 0.1 mL of LysCP28 was dropped evenly on the surface of the meat pieces.

  1. Lien 196: Why 1:5 dilution was made instead of 1:10?? And how the number of bacteria were estimated then?

Answer: Line 225, we have revised to 1:10 and diluted concentration of bacteria was 103 CFU/g. in manuscript. According to pervious research reference 25 “Tian, Y., Wu, L., Lu, R., Bao, H., Zhou, Y., Pang, M., Brown, J., Wang, J., Wang, R., Zhang, H. Virulent phage vB_CpeP_HN02 inhibits Clostridium perfringens on the surface of the chicken meat. Int J Food Microbiol. 2022, 363, 109514”. The number of bacteria were estimated by the dilution gradient multiplied by the number of bacteria harvested and enumerated on TSC and SFP media in manuscript method “2.7. Lytic activity in duck meat” line 226-229.

  1. Line 223: 2.3: please take the methodology part of this section to the methodology part. Don’t put the methods in the results part.

Answer: Line 135-137, the methods has been moved to the methodology part “2.3. Determination of the activity spectrum using plate lysis assay” in manuscript.

  1. Some scientific name of not italic.

Answer: Thank you for your suggestion. The scientific name has been revised in the manuscript.

  1. Some experiments are discussed in the results but missing in the methodology part.

Please read the manuscript thoroughly and correct write the missing methods.

Answer: Thank you for your suggestion. All methods have been added in the manuscript.

  1. Methods detail of 3.1 of the results section is missing in methodology.

Answer: Line 116-125, methods of 3.1 has added in manuscript “2.2 Cloning, expression, and purification of endolysin LysCP28”.

  1. Methodology of 3.2, 3,4 is missing

Answer: Line 134-146, methodology of 3.2 has been added in manuscript “2.3. Determination of the activity spectrum using plate lysis assay”.

Line 147-168, methodology of 3.4 has been added in manuscript “2.4. Time-kill assay and stability of LysCP28”.

  1. There are some formatting mistakes in the text of the manuscript body

Answer: Thank you for your suggestion. The formatting mistakes have been revised.

  1. Conclusion is missing??

Answer: Thank you for your suggestion. Line 438-449, the conclusion has been added in manuscript.

Round 2

Reviewer 2 Report

The manuscript need a thorough read to omit gramatical errors.

For example, I have made a correction in the manuscript file, Line 355-360.

"A low dose of LysCP28 (5 µg/mL) was not as effective in removing C. perfringens as the high dose, an approximately 1.89-log reduction of C. perfringens was still observed after 72 h of treatment. However, the density of C. perfringens in control sample was still close to the initial concentration (> 90%) after 72 h of incubation".